# The Special Developmental Biology of Craniofacial Tissues Enables the Understanding of Oral and Maxillofacial Physiology and Diseases

**DOI:** 10.3390/ijms22031315

**Published:** 2021-01-28

**Authors:** Manuel Weber, Falk Wehrhan, James Deschner, Janina Sander, Jutta Ries, Tobias Möst, Aline Bozec, Lina Gölz, Marco Kesting, Rainer Lutz

**Affiliations:** 1Department of Oral and Maxillofacial Surgery, Friedrich-Alexander University Erlangen-Nürnberg (FAU), 91054 Erlangen, Germany; falk.wehrhan@outlook.com (F.W.); jutta.ries@uk-erlangen.de (J.R.); tobias.moest@uk-erlangen.de (T.M.); marco.kesting@uk-erlangen.de (M.K.); rainer.lutz@uk-erlangen.de (R.L.); 2Private Office for Maxillofacial Surgery, 91781 Weißenburg, Germany; 3Department of Periodontology and Operative Dentistry, University of Mainz, 55131 Mainz, Germany; james.deschner@uni-mainz.de; 4Private Office for Oral Surgery, 96049 Bamberg, Germany; sander.janina@googlemail.com; 5Department of Rheumatology and Immunology, Friedrich-Alexander University Erlangen-Nürnberg (FAU), 91054 Erlangen, Germany; aline.bozec@uk-erlangen.de; 6Department of Orthodontics, Friedrich-Alexander University Erlangen-Nürnberg (FAU), 91054 Erlangen, Germany; lina.goelz@uk-erlangen.de

**Keywords:** maxillofacial development, cranial neural crest, developmental biology, orthodontics

## Abstract

Maxillofacial hard tissues have several differences compared to bones of other localizations of the human body. These could be due to the different embryological development of the jaw bones compared to the extracranial skeleton. In particular, the immigration of neuroectodermally differentiated cells of the cranial neural crest (CNC) plays an important role. These cells differ from the mesenchymal structures of the extracranial skeleton. In the ontogenesis of the jaw bones, the development via the intermediate stage of the pharyngeal arches is another special developmental feature. The aim of this review was to illustrate how the development of maxillofacial hard tissues occurs via the cranial neural crest and pharyngeal arches, and what significance this could have for relevant pathologies in maxillofacial surgery, dentistry and orthodontic therapy. The pathogenesis of various growth anomalies and certain syndromes will also be discussed.

## 1. Introduction

### 1.1. The Body Plan

The Hox genes define specific morphological characteristics along the body axis from simple insects to complex mammals. Remarkably, the order of the different Hox genes on a chromosome corresponds to the order of their expression in successive body segments. If one Hox gene fails, the affected body region takes over characteristics of the neighboring Hox genes. For example, in the fruit fly (Drosophila), an additional pair of wings or in the mouse, an additional pair of ribs could be induced. While the Hox genes in Drosophila are still located on one chromosome, in humans the Hox genes are distributed in four clusters on four chromosomes. Besides the Hox genes, there are numerous other genes that code and regulate the three-dimensional body plan. These include genes from the gene families Pax, T-Box, Wnt and Sonic Hedgehog [1].

### 1.2. Early Embryonic Development

After fertilization of the ovum and formation of the zygote, cleavage divisions initially occur without the total volume of cytoplasm increasing. The resulting morula then transforms into the blastocyst. The outer cells of the blastocyst form the trophoblast, while the cells inside are called embryoblast [2]. The trophoblast later forms the placenta, while the embryoblast represents the later embryo [1]. The embryoblast forms two cell layers: epiblasts and hypoblasts. The epiblast develops the amniotic cavity and the hypoblast develops the yolk sac. The contact surface of the epiblast and hypoblast is called the germinal disc, which produces the embryonic cotyledons in the further course of development [1,3].

### 1.3. Development of the Cotyledons

The development of the three embryonic cotyledons begins with the immigration of cells into the space between epiblast and hypoblast. This process begins with the formation of the primitive gutter (primitive strip) on the epiblast. There, proliferation and epithelial–mesenchymal translation (EMT) of cells of the epiblast occurs. EMT gives the cells migratory abilities and allows them to enter the gap between epiblast and hypoblast. The cells that come in contact with the yolk sac underneath the epiblast displace the hypoblast cells and form the definitive entoderm [1,3]. The cells that migrate laterally between epiblast and entoderm form the mesoderm. After differentiation of the mesoderm, the overlying tissue is called the ectoderm. The development of the three cotyledons is called gastrulation [1,2].

### 1.4. Development of the Neural Tube

At the anterior pole of the primitive gutter a thickening, the so-called primitive node, occurs. Behind this, a depression is formed, from which the chorda dorsalis emerges in further development [1]. The chorda dorsalis does not form any embryonic tissue itself, but only induces the development of the nervous system by inducing neural tube formation. The neural tube folds out of the ectoderm and then comes to rest underneath it. In the process of unfolding the neural tube from the ectoderm, a lateral migration of cells creates the neural crest [1].

The neural crest (NC) is a multipotent embryonic cell population with stem cell characteristics that undergoes extensive migration during embryogenesis and can produce a variety of tissues such as neurons, melanocytes, cartilage and bone [4].

Cranial, vagal, stem and sacral neural crest cells can be distinguished, which are characterized by different migration pathways and differentiation into different target tissues [4]. Cranial neural crest (CNC) cells are of particular importance for the understanding of craniofacial development.

### 1.5. Development of the Head

The formation of the head is, in essential aspects, different from the formation of the tissues in the rest of the body. For example, practically all connective tissue (cartilage, bone, fibroblasts) of the craniofacial region originates from the neural crest, whereas the connective tissue in the rest of the body is of mesodermal origin. The neural crest can therefore form not only nerve tissue in the head as in the rest of the body, but also mesenchymal tissue [1].

#### 1.5.1. Development of the Pharyngeal Arches

While with the water-living vertebrates, the pharyngeal arches are the origin for the development of the respiratory system, the pharyngeal arches undergo a functional change with the land-living vertebrates. The terrestrial vertebrates develop a new respiratory organ—the lung—from the gullet [1]. In the region of the pharynx, the pharyngeal arches are formed by proliferation of cells migrating from the neural crest. These are five—a sixth is only rudimentary—clasp-shaped prominences, each containing a vessel, a nerve branch and a muscle segment. These pharyngeal arches are separated from each other by pharyngeal furrows on the outside (ectodermal) and by so-called pharyngeal pouches on the inside (entodermal) [3].

From the first pharyngeal arch, among other things, the masticatory muscles develop, as well as the N. mandibularis of the N. trigeminus, the Meckel cartilage, which is involved in the formation of the lower jaw, as well as the upper jaw and lower jaw prominence. The second pharyngeal arch develops mainly the mimic muscles and the facial nerve, as well as the Reichert cartilage. The third pharyngeal arch is responsible, among other things, for the upper muscles of the pharynx and the N. glossopharyngeus, while the fourth is responsible for the muscles of the lower pharynx and the N. vagus. fifth and sixth pharyngeal arches are sometimes involved in the development of the internal laryngeal muscles [1].

#### 1.5.2. Development of the Face

At the beginning of the face-development, the mouth-bay (Stomatodeum) is framed by five facial prominences. The five facial prominences—the unpaired frontal nose prominence, the paired maxillary prominences and both mandibular prominences—are formed by proliferation of cranial neural crest cells. The frontal prominence can be divided into a medial and lateral nasal prominences, which enclose the olfactory pits. In regular facial development, the medial nasal process merges with the maxillary ridges on both sides. Failure to achieve this fusion results in cleft malformations of the lip and jaw [3]. The so-called primary palate is formed by fusion of the two medial nasal prominences with each other. This forms the os incisivum in further development. The secondary palate, on the other hand, is formed by fusion of the palatal processes of the two upper jaw prominences. If the fusion does not occur, cleft palates result [3].

#### 1.5.3. Development of the Tongue

The development of the tongue begins with the anterior growth and fusion of the lateral tongue prominences, which originate from the first pharyngeal arch. Dorsally, the unpaired tuberculum impar follows the tongue prominences. The root of the tongue dorsally of the sulcus terminalis, on the other hand, is formed by parts of the second, third and fourth pharyngeal arches [1,3]. The development from the first four pharyngeal arches explains the innervation of the tongue. The sensitive innervation in the anterior two thirds (until the sulcus terminalis) is performed by the lingual nerve from the mandibular nerve (first pharyngeal arch). The pharyngeal part of the tongue is innervated by the glossopharyngeal nerve (third pharyngeal arch) and the superior laryngeal nerve originating from the vagal nerve (fourth pharyngeal arch). In the front, two-thirds of the sensory (taste) innervation of the tongue is performed by the Chorda tympani from the facial nerve (second pharyngeal arch). The pharyngeal innervation for taste corresponds to the described sensitive innervation [1,3].

#### 1.5.4. Development of the Nervous System in the Head Area

As described above, the so-called neural tube is created during neurulation. During further embryogenesis, the brain emerges from the front two-thirds of the neural tube, whereas the rear third becomes the spinal cord. In the cranial neural tube, curvatures occur and three functionally different sections are formed: the forebrain (prosencephalon), the midbrain (mesencephalon) and the rhombencephalon [1]. In the area of the forebrain, there are vesicular prominences, from which the paired cerebral hemispheres (telencephalon) develop. This leads to a division of the forebrain into the unpaired, central diencephalon and the paired hemispheres of the cerebrum. The cerebellum develops from the roof of the rhombic brain [1].

## 2. Methods

For the preparation of this review, a systematic literature research was conducted. For this purpose, the most well-known German and English textbooks on embryology were used. In addition, a search was conducted using the Pubmed database (https://www.ncbi.nlm.nih.gov/pubmed/). The articles were first screened by title and ab-stract. If the title and abstract were suitable, the full texts were downloaded as PDF. Sys-tematic reviews and original papers were included. If available, current literature was used.

The electronic search in Pubmed initially used the following search terms:

Cranial neural crest;Head and neck development;Craniofacial Abnormalities;Branchial arch;Jaw development;Tongue development;Mandibular osteogenesis;Cleft palate.

Original works, case series and review works were taken into account. Work that was an isolated case report was excluded. Where available, papers published after 2000 were used.

## 3. Special Features of Craniofacial Development

### 3.1. The Role of the CNC

Three-quarters of human malformations affect the craniofacial region [5]. This fact highlights the complexity and the susceptibility of the embryological processes for the formation of craniofacial tissue.

The craniofacial tissues are mainly derived from cells of the cranial neural crest. These cells develop in the dorsal region of the neural tube and then migrate into the facial prominences and the 1st to 4th pharyngeal arches [5].

In further development, they contribute to the formation of neuronal, skeletal, dermal and mesenchymal structures [5].

CNC cells interact with other cells of the craniofacial tissues in many different ways during their migration, but also after completion of morphogenesis [5].

The skeleton of the face and a large majority of craniofacial connective tissue is derived exclusively from cells of the cranial neural crest [5]. These are pluripotent cells with exceptional migratory capabilities [6].

#### 3.1.1. Creation of CNC Cells

The CNC cells are formed in the border region between neural and non-neural ectoderm dorsal to the neural tube [5]. How exactly the differentiation of CNC cells is initiated and regulated is not yet fully understood. It is assumed that the WNT- and bone morphogenetic protein (BMP)-signaling pathways are of particular importance [5]. The formation of neural crest cells occurs during embryogenesis at about the time of neural tube closure.

The so-called EMT is required to initiate the CNC. The EMT is a prerequisite for the migration capabilities of the CNC [5,6].

Through EMT, epithelial cells can leave their tissue network and migrate to other regions of the organism. Besides the physiological importance of EMT in embryogenesis, EMT plays an important pathophysiological role in the invasion and metastasis of malignant tumors [5].

For EMT, the CNC cells must first lose their apico–basal polarity and degrade intercellular adhesion molecules, such as cadherins and tight junctions [5].

At the transcriptional level, EMT is mainly regulated by the transcription factors Snail1 and Snail2 (slug) [5].

#### 3.1.2. Migration of the CNC Cells

CNC cells, which originate in the forebrain and rostral midbrain, migrate to the frontonasal and periocular facial region. CNC cells from the caudal midbrain migrate to the maxillary portion of the first pharyngeal arch. In the rhombic brain, CNC cells are derived from the seven rhombomeres and migrate into the pharyngeal arches [6] (Figure 1).

During their migration, the CNCs move along defined routes. The migration begins in a continuous wave and then splits into three separate streams [6].

The migration of CNC cells into the craniofacial tissues is regulated by different cytokines. The exact regulation of these cytokines is still unknown. However, it is known that there are attractive and repellent signal molecules [5]. Furthermore, CNC cells show contact inhibition of movement. Thus, the movement of a larger group of migrating CNC cells can be directed in one direction [6]. CNC cells inside the migrating cell group are thus prevented from disordered movement, while the cells at the tip of the cell assembly do not experience contact inhibition of movement when moving forward [6].

The CNC cells and their migration seem to be of crucial importance for the individual face shape. Transplantation experiments have shown that the final face shape in a host embryo is determined by the donor’s CNC cells [7]. The transplantation of mouse CNC cells into chicken embryos has led to the development of dentate jaws. These experiments show that the CNC cells are able to activate the genetic programs for tooth development in the ectodermal cells of the chicken [7]. CNC transplantations between duck and quail showed that the shape of the cranial feathers matched the profile of the CNC donor. Even in higher mammals, the CNC cells influence both skeletal and soft tissue facial shape [7].

After the CNC cells have arrived in their target region, they must differentiate on site. It is not clear whether different CNC cells already carry the information for their final differentiation intrinsically, or whether local signals in the target area are responsible for their differentiation [5].

The CNC cells maintain their multipotent status until late embryonic development [5].

The peripheral nerves are also important for the later embryonic development of the craniofacial tissues. Nerve-associated CNC cells play a special role here, which can differentiate themselves from other cell types and influence craniofacial morphogenesis [7]. Thus, peripheral nerves can be considered as stem cell niches, from which different cell types, such as bone marrow mesenchymal cells and melanocytes, can differentiate. In addition to their role as pigment cells, melanocytes play a decisive role in the development of the inner ear, where they contribute to the survival of sensory hair cells.

In the dental pulp, CNC-derived cells of the pulp nerves play an important role in the regeneration of mesenchymal pulp cells and odontoblasts [7].

#### 3.1.3. Contribution of the CNC Cells to the Development of Facial Prominences

The embryonic face consists of the unpaired forehead-nose prominence, the paired maxillary and mandibular prominences.

The forehead, nose, upper lip, philtrum and primary palate are formed from the frontonasal prominence (FNP). For the proper development of the FNP, interaction between the migrating CNC cells with the local epithelial cells of the facial ectoderm and the cells of the forebrain is necessary [5]. The lateral region of the FNP fuses with the lateral nasal prominence and the maxillary prominence. Maxillary and mandibular prominence originate from the first pharyngeal arch. Their development requires the interaction of the immigrating CNC cells with the local cells of the surface ectoderm, mesoderm and pharyngeal entoderm [5]. In the pharyngeal arches, a characteristic spatial arrangement of immigrating CNC and the local cells of the three primary cotyledons occurs. Thus, the mesodermal cells are located in the center of the pharyngeal arches and are surrounded by the CNCs [5]. The outer closure is formed by epithelial cells of the ectoderm and the inner closure by the entodermal epithelial cells of the pharynx [5].

The CNC decisively controls the morphogenesis of the face from different cell populations. During the embryonic development of the face, bone emerges from the CNC cells, while muscle develops from the mesodermal cells [5]. Signals from the CNCs control the differentiation of mesodermal cells into myoblast progenitor cells and, subsequently, the organization of these cells around the developing skeletal elements [5].

Disturbances in migration and growth of CNC cells are of particular importance for the pathogenesis of cleft malformations [5]. Cleft malformations are the most important congenital craniofacial malformations, occurring in one patient per 700 births and requiring complex combined surgical and orthodontic treatment procedures.

While the primary palate originates from the FNP, the secondary palate develops from the palatal processes of the maxillary prominence [5]. The palatal processes consist of CNC cells surrounded by epithelial cells [5]. The Wnt signaling pathway is of particular importance for sufficient growth of CNC cells in the maxillary process. Reduced activation of the Wnt signaling pathway can lead to reduced growth and, thus, to cleft deformities of the palate. After the palatal processes have approached, the epithelial cells must be removed to allow the CNC to fuse. This can be achieved by apoptotic cell death or by migration of the epithelial cells [5]. The epidermal growth factor receptor (EGFR) pathway probably plays an important role in this process [5].

Similar to haematological stem cells, CNC cells initially show pluripotency while they are increasingly restricted in their developmental potential during further embryogenesis. However, it is not yet clear what proportion of pluripotency is retained by CNC cells into adulthood [5].

Most of the teeth are formed by CNC cells. Thus, the dentin, cementum, periodontal ligament and most of the pulp are made of CNC. Only the blood vessels of the pulp and enamel are not CNC derivatives [6] (Figure 1).

#### 3.1.4. Cellular Characteristics of CNC Cells

CNC cells differ in their development potential from other NC cells of the body strain. CNC cells, for example, activate genes of chondral differentiation [6].

At the transcriptional level, the transcription factors Sox10, Sox9 and Ets1 are characteristic for CNC cells and play a major role in the regulation of their effector genes [6].

#### 3.1.5. The Role of CNC Cells in Tooth Development

Interactions of epithelial and mesenchymal cells are crucial for tooth development [8]. Tooth development begins when cells of the oral epithelium send signals to the underlying mesenchymal tissue derived from CNC cells. During tooth development, the epithelial cells differentiate into ameloblasts, while the CNC mesenchymal cells form odontoblasts [8].

However, the exact signaling pathways that regulate the formation of the tooth roots or the number of roots of each tooth are still unknown [8].

It is known that the Hertwig’s epithelial sheath (HES) develops apically from the crown of the tooth.

### 3.2. Determination of the Body Axes by Hox and Dlx Genes

Embryologically, the segmental body structure is regulated by the Homeobox (Hox) genes [9]. The HOX genes are phylogenetically strongly conserved. In addition to all animals, the blueprint of plants and fungi is regulated by the Hox genes. In mammals the Hox genes are organized in four clusters (Hox A to D) [10]. Each cluster consists of 9 to 11 Hox genes. During early embryogenesis the Hox genes control the development of the body along the longitudinal axis [10].

In contrast to the rest of the body, Hox gene expression is absent in CNC cells of rhombomeres 1 and 2 [11]. The CNC cells of the first two rhomboids migrate into the first pharyngeal arch and form the structures of the neurocranium and the jaws [12]. The Hox positive CNC cells of the caudal rhombomers (r3 and below) form the cartilages of the larynx [12].

Hox-negative CNC cells are regulated in their morphogenesis by distal-less (Dlx) genes [11]. The Dlx code provides the CNC cells with structural information and regulates their polarity within the pharyngeal arches along the dorsal–ventral and proximal–distal axis. Dlx1 and Dlx2 are expressed in both the maxillary and mandibular prominence in the first pharyngeal arch, while Dlx5 and Dlx6 are only expressed in the mandibular prominence. In contrast, Dlx3 and Dlx4 are restricted there [11]. Thus, the Dlx combination code regulates the differentiation of the CNC cells in the first pharyngeal arch into maxilla and mandible. The Dlx code 1/2 defines the cells of the maxilla and the expression of Dlx 1/2/5/6 defines the mandible [11]

Besides Dlx, many other genes are involved in the morphogenesis of CNC derivatives. Signals from the fibroblast growth factor (FGF) family have an important influence on the formation of the rostral–caudal axis [11].

The proximal–distal axis is mainly determined by FGF and BMP. Due to the action of the growth and differentiation factors, FGF and BMP, the expression of the transcription factors Barx1 and Dlx2 is restricted to the proximal part of the first pharyngeal arch, while Msx1, Msx2 and Alx4 are restricted to the distal part [11]. Among other things, the transcription factor, Barx1, is involved in regulating the morphogenesis of teeth from incisors to molars [11].

### 3.3. Development of the Jawbone

Meckel’s cartilage is a hyaline cartilage, which serves as a guiding structure for the development of the mandible bone during embryogenesis. The development of Meckel’s cartilage within the mandibular prominence begins with the condensation of CNC cells in the area of the later first molar [11]. These cells then differentiate into chondrocytes and form the rod-shaped Meckel cartilage. The Meckel’s cartilage initially extends in a ventromedial and dorsolateral direction on both sides and fuses at the most distal tip in the area of the later symphysis mandibulae. The proximal sections of Meckel’s cartilage change shape to develop the hammer and anvil bone of the middle ear [11].

The exact regulation of the formation of Meckel’s cartilage at the transcriptional level is still unknown. However, the chondrogenic transcription factor Sox9 plays an important role. Sox9 knockout mice cannot form Meckel cartilage. However, despite the absence of Meckel cartilage, these animals develop a reduced mandible bone. This shows that the Meckel cartilage is not necessary for initiating mandibular development [11].

The different parts of the lower jaw show different ossification. The distal portion of the mandible is enchondrally ossified from the symphysis [11]. In the middle part of the corpus mandibulae, intermembranous ossification occurs, while enchondral ossification occurs again in the proximal part. In intermembranous ossification, the CNC cells condense and then differentiate into osteoblasts. These cells then begin to secrete osteoid, which then calcifies secondarily. The differentiation of osteoblasts is regulated by various transcription factors. Dlx5 induces the expression of Runx2. Runx2 in turn induces the expression of Osterix, which regulates the differentiation of preosteoblasts into mature osteoblasts [11]. Furthermore, ossification is induced by BMPs [13]. The late differentiation of osteoblasts finalize in osteocytes embedded in calcified bone, this step is dependent on MEPE and Dmp1 factors.

In enchondral ossification, the bone is formed using a cartilaginous template. This leads to a condensation of the CNC cells, which then differentiate into chondrocytes. The differentiation of the osteoblasts begins first in the perichondrium and then progresses centrally. The differentiation of osteoblasts is controlled by signaling pathways such as IHH, NOTCH, WNT and BMP, and by the transcription factors Dlx5, Runx2 and Osterix [11].

### 3.4. Development of the Tongue

The tongue and the lower jaw have a common development—biological origin. They originate simultaneously from the mandibular prominence and their development is closely coordinated. In the medial part of the mandibular prominence, the tongue protrusion is formed, which also consists of CNC cells. This leads to the immigration of myoblasts from the occipital somites [11]. Thus, the connective tissue and blood vessels originate from the CNC, while the tongue muscles are formed from mesenchymal myoblasts. The CNC cells are important for the initiation and regulation of tongue development. Thus, the CNC cells can be understood as a matrix for the migrating myoblasts and they determine the pattern of muscle development. Furthermore, the CNC cells regulate proliferation and differentiation of the myoblasts. The Dlx genes 5 and 6 play an important role in this process. A loss of function of the Dlx5/6 genes in CNC cells leads to the absence of masticatory muscles and disturbed tongue development [11].

In addition, the Hedgehog signaling pathway plays an important role in the development of the tongue. CNC cells react to Hedgehog activation of cells of the tongue epithelium and influence the development of myoblasts. In addition, it has been shown that disturbances in the transforming growth factor (TGF) beta signaling pathway also lead to a defective tongue development [11].

## 4. Clinical Impact of the Craniofacial Development

### 4.1. Significance for Specific Diseases of Craniofacial Tissue

Craniofacial tissue behaves biologically differently compared to extracranial tissue. One possible cause is the origin of craniofacial bone/tissue from the cranial neural crest (Figure 2). Cranial neural crest-based tissues seems to be characterized by the biological peculiarity that cyclic forces evoke greater anabolic responses of craniofacial sutures as well as cranial base cartilage since gene expression, cell proliferation, differentiation and matrix synthesis were mechanically regulated [14]. Mechanical force thus influences genetics, whereby the onset of temporomandibular disorders can be explained down to the genetic level [14,15]. The response of CNC-derivates to forces is relevant for orthodontic treatment, with the aim to modulate growth. For example, in class 2 deformities (mandibular retrognathy), a stimulation of the condylar growth is desirable. There are several orthodontic approaches to achieve this goal in the growth periods of children. These approaches have in common an increase of muscular activity, repositioning the mandible anteriorly and a relief of compressive forces [14,15]. Another example of clinical relevance of cranial neural crest-dependent diseases is the differences between craniofacial and extracranial osteosarcoma

Early metastasis is characteristic of extracranial osteosarcoma. In contrast, craniofacial osteosarcoma rarely develop metastasis [16,17]. In addition, craniofacial and extracranial osteosarcomas show a different clinical prognosis. Craniofacial osteosarcomas have a five-year survival rate of about 77%, while extracranial osteosarcomas have a five-year survival rate of only 55% [18]. The overriding clinical problem with craniofacial osteosarcoma is frequent tumor recurrence. This may be due to the difficulty of safe R0 tumor resection due to the close proximity to vital anatomical structures [16]. The different clinical behavior of craniofacial versus extracranial osteosarcoma may be due to the different developmental biology of craniofacial and extracranial bone. A different activation of the Hedgehog signaling pathway and also possible immunologic differences between craniofacial and extracranial osteosarcomas have already been shown [19].

The jawbone exhibits increased resistance to osteoporosis compared to the extracranial bone [20]. There is also a disease that affects almost exclusively the jaw—drug-associated necrosis of the jaw MRONJ. This is caused by antiresorptive drugs such as bisphosphonates or denosumab, as well as angiogenesis inhibitors or various tyrosine kinase inhibitors. Although the aetiopathogenesis of this disease is not yet fully understood, it is believed that the developmental characteristics of the jawbone play an important role in the occurrence of the disease [21,22], principally through the strong activation of the bone resorbing cells, osteoclasts.

### 4.2. Significance for Orthodontic Treatment

The orthodontist takes care of the physiological development of craniofacial growth and occlusion by preventing oral dysfunction, regulating jaw growth and moving the teeth within the alveolar bone when necessary, thereby using the special features of the craniofacial tissues. The alveolar bone, for example, is special since it is inducible by orthodontic tooth movement [23].

These tooth movements not only cause a change in occlusion, but also model the maxillofacial hard tissue and, as a result, the soft tissue. It must be highlighted, that the outcome of dentofacial orthopedic appliances is mostly due to deflection/bending of the alveolar bone and of remodeling processes of the periodontal tissues instead of skeletal increase due to growth stimulation. In this context, mechanical stimuli seem to play an essential role for cell differentiation, proliferation and metabolism due to regulation of expression of transcription factors, cytokins, growth factors, enzymes and structural proteins [24]. Functional orthodontic and extraoral appliances, in particular, take advantage of the physiological growth of the maxillofacial structures and modulate this growth. In addition, the orthodontist is regularly confronted with various malformations of the maxillofacial tissue. In the case of cleft malformations, for example, protracted orthodontic treatment is required to accompany the surgical interventions.

Furthermore, differences in the development of cells of the mucosa compared to cells of the skin enable the understanding of orally induced tolerance against nickel via orthodontic treatment, as well as the mechanism of sublingual immunotherapy [25,26].

Another clinical observation that should be considered in orthodontic treatment planning is that the craniofacial bone exhibits faster bone healing and increased remodeling, which is crucial for a targeted and successful orthodontic therapy [20]. Therefore, diseases or medications affecting craniofacial development or bone remodeling impair orthodontic treatment favoring maxillofacial malformation and malocclusion. This may cause difficulties in mastication and speech, favor craniomandibular disorders and lead to a reduced quality of life [27].

Hence, in order to be able to optimally support the developmental processes, dentists and orthodontists should know that the human skull could be differentiated in the viscerocranium and neurocranium, which significantly differ in their development and growth and what special features play a role in these processes [28].

### 4.3. Impact for Oral Implant Osseointegration and Mesoderm Derived Bone Transplants

Branemark’s discovery of titanium osseointegration took place on titanium implants for intravital microscopy of the rabbit fibula [29]. Since then, much of what we have learned about osseointegration in the past decades has been done in studies on the long tubular bones of experimental animals, as these are easily accessible and have a large osteogenic marrow space [30,31]. In everyday clinical practice, most dental implants are inserted into the jawbone. Exceptions are patients who have had reconstruction of the maxilla or, more frequently, the mandible with a microvascular fibula graft, e.g., due to tumor disease of the oral cavity. In these patients, the dental implants are inserted into a long bone of mesodermal origin and can be directly compared to those placed into the jaw bone. Wijbenga et al. showed an implant survival rate of 95% in a follow-up period of 0 to 155 months after microvascular fibular reconstruction and dental rehabilitation with endosseous implants [32]. However, current data on the functional outcome and quality of life of patients are limited due to their study. Similar implant survival rates were found by Sozzi et al., who found an implant survival rate of 98% after 7.8 years following microvascular reconstruction of the jaws. No statistically significant differences were found between maxillary and mandibular or in irradiated and nonirradiated patients [33]. The implant survival rate was similar to that of Howe et al., who, in a meta-analysis, estimated the 10-year survival of dental implants in the jawbone of healthy patients to be 96.4% [34]. In our research group, we were able to show that porcine calvarial frontal bone (neural crest-derived dermatocranium) can serve as a model for bone regeneration of the human maxilla (neural crest-derived splanchnocranium) [35,36]. In this model, we were also able to examine different implant surface modifications and local gene therapy to improve osseointegration of dental implants [37,38,39,40].

Mouarett et al. found different rates of bone regeneration in a mouse model, when comparing the bony healing of defects in the tibial bone vs. defects in the maxillary bone [30]. They also found an influence of the maxillary periosteum on implant osseointegration of the murine maxilla. This is in good agreement with our own data, as we were able to demonstrate supracortical peri-implant bone formation through periosteal elevation in an established model of the porcine frontal skull [41]. When grafting bone into the jaw area, in addition to homotopic grafts from the jaw bone itself, heterotopic bone components of mesodermal origin (fibula, scapula, iliac crest or parietal cranial calvaria) are frequently transplanted into the jaws [36]. Cells of neurocrestal origin, as well as cells of mesodermal origin, can ossify intramembranously as well as endochondrally [36]. The unanswered question is what happens in detail to the transplanted cells? What is the influence of the embryological origin of the cells and what is the effect of the bony environment in the jawbone? A more precise understanding of these processes could contribute decisively to optimizing the regenerative possibilities of the transplanted bone tissue [33]. This would not only contribute to the well-being of patients requiring bone transplants, but would also have an immense economic benefit, as bone is the second most frequently transplanted tissue after blood [42].

### 4.4. Impact for Syndromes and Malformations

Malformations of the derivatives of the CNC account for about one third to one half of all congenital malformations in humans [6]. The clinical significance of the CNC will be illustrated below using a few syndromes and malformations as examples.

#### 4.4.1. Fetal Alcohol Syndrome

Fetal alcohol syndrome (FAS) is the most common teratogenic stress in humans and the best-known cause of developmental disorders. FAS is characterized by lifelong behavioral and cognitive deficits, as well as impaired attention, learning and motor skills [43]. Phenotypically, affected individuals can be recognized by craniofacial dysmorphia. These include small eyelid crevices, a flattened philtrum and a thin upper lip, as well as micrognathia and reduced interocular distance. Micrognathia is often accompanied by tooth displacement and impactations that require orthodontic and surgical treatment. Direct toxic effects of ethanol on the migrating cells of the CNC are considered to be the cause of these changes, whereas the neural crest of the strain does not seem to be affected [43]. In FAS, for example, no impairment of the autonomic nervous system or of the melanocytes derived from the neural crest of the strain is observed. The exact causes for the CNC specificity of ethanol toxicity are not known [43]. In animal experiments, ethanol led to a disturbed induction of CNC formation, impaired migration of CNC cells with increased apoptosis and, consequently, to morphological changes in the craniofacial structures of embryos. A mechanism of ethanol action on CNC cells is mediated by the Hedgehog signaling pathway. Ethanol leads to an impairment of the formation of the ligand Sonic Hedgehog [43].

#### 4.4.2. Treacher Collins Syndrome

Treacher Collins Syndrome (TCS) is a congenital disorder of craniofacial development. TCS is also known as Dysostosis mandibulofacialis. Characteristics of TCS include hypoplasia of the facial bones, particularly of the upper jaw, lower jaw and zygomatic complex [44]. In severe cases, the zygomatic arches may be completely absent and cleft malformations may be present. Jaw hypoplasia often leads to malocclusion with an anterior open bite. In addition to hypodontia, there are also changes in the shape of the teeth [44]. Moreover, malformations of the outer ears with atresia of the auditory canal and anomalies of the middle ear bones are common, which consequently lead to hearing disorders. In addition, there are impairments in brain development, mental retardation and psychomotor retardation. Micrognathia can cause airway obstruction caused by the tongue directly postpartum [44].

The molecular pathomechanism of TCS is now relatively well understood. In affected individuals, a mutation of the TCOF1 gene is present. The gene product of TCOF1 is jointly responsible for the initiation, proliferation and survival of CNC cells. In animal experiments, mutations of TCOF1 lead to a reduced number of CNC cells with undisturbed migration abilities of the CNC cells [44].

#### 4.4.3. Cleft Malformations

Despite numerous genome-wide analyses, the evidence of a clear genetic cause for cleft malformations is limited [45]. In this context, missense-mutations in the IRF6-gen as well as in the Grainy-head-like-3 (GRHL3)-gen could be detected as casual risk factors [46,47,48].

As described above, migration processes of CNC cells and their interaction with local cell populations play an important role in the pathogenesis of cleft malformations [5]. For example, correct proliferation of mesenchymally differentiated CNC cells is required for upper lip closure. Recent studies show a possible important role of the Hedgehog signaling pathway [45]. The Hedgehog signaling pathway is an important control element of epithelial–mesenchymal interactions during orofacial development. In animal experiments it could be shown that the formation of cleft lips is associated with a reduced proliferation of CNC cells of the medial nasal processes [45]. A reduced expression of the ligand Sonic Hedgehog led to a reduced gene expression of the transcription factor Foxf2 in the downstream of the signaling pathway. By increasing the expression of Sonic Hedgehog or Foxf2, an increased CNC proliferation could be achieved and thus the development of cleft lip could be counteracted [45]. Nevertheless, the various forms of cleft lip malformations in humans are a highly complex group of multifactorial malformations in which various genetic and environmental factors interact [45].

#### 4.4.4. Pierre Robin Sequence

The Pierre Robin Sequence (PRS) is characterized by a small lower jaw (micrognathia), a posterior displacement of the tongue (glossoptosis) and the associated obstruction of the upper airways. In addition, there is usually a cleft palate and bimaxillary retrognathia with reduced sagittal length of the mandible and maxilla [49]. The exact genetic mechanism of PRS is still unknown. A disturbed migration of CNC cells into the first two pharyngeal arches is assumed [49,50]. Mutations of Sox9 and of the bone morphogenic protein (BMP) signaling pathway are discussed as potential causes [51].

#### 4.4.5. Hemifacial Microsomia

Hemifacial microsomia is characterized by disorders of the development of the upper jaw, lower jaw, outer ear and middle ear, as well as of the trigeminal and facial nerve on the affected side of the face [52]. Cardiac, vertebral and central nervous malformations are also possible. The phenotypic expression of these developmental disorders is highly variable [49]. Causes are often discussed as disturbed blood flow during the morphogenesis of craniofacial tissues or localized ischemia. Besides such environmental factors, genetic influences are also suspected [52]. Thus, hemifacial microsomia can be regarded developmentally as a malformation of the first two pharyngeal arches. Thus, a disturbed morphogenesis of CNC derivatives is present. Mutation analyses in affected patients have revealed changes in various genes involved in the development and vascularization of CNC cells. Nevertheless, hemifacial microsomia seems to be a heterogeneous, multifactorial disease pattern [52].

#### 4.4.6. Goldenhar Syndrome

Goldenhar syndrome—also known as oculo-auriculo-vertebral syndrome—can be understood as an extended spectrum of hemifacial microsomia. The malformation complex is characterized by impaired development of the eyes, ears, lips, tongue, palate, jaw, zygomatic bone and dental deformities. It is caused by a malformation of the first and second pharyngeal arches [53]. In addition, ocular dermoid cysts, spinal anomalies and malformations of internal organs, such as the heart and kidney, can occur to varying degrees. Although various genetic changes have been detected in patients with Goldenhar syndrome, no clear genetic cause has been identified. A combination of genetic and environmental factors is probably pathogenetically relevant. For example, it has been discussed that abnormal development of vascularization in the fourth week of pregnancy—when the first two pharyngeal arches develop—could be the cause [53].

## 5. Conclusions

Maxillofacial tissues are characterized by an embryology unique in the human organism. This explains many of the peculiarities of maxillofacial tissues, such as increased bone regeneration and good modulation ability, but also the occurrence of malformations. The complex development via cranial neural crest and pharyngeal arches, as well as the involvement of complex cell migration and multiple redifferentiations, can explain the relatively frequent occurrence of craniofacial malformations.

## Figures and Tables

**Figure 1 ijms-22-01315-f001:**
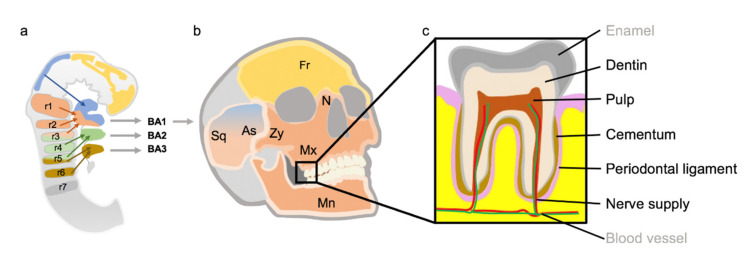
Migration, skeletal and dental derivatives of cranial neural crest (CNC) cells. (**a**) Colonization of the pharyngeal arches by CNC cells from forebrain, midbrain and rhombic brain. In the rhombic brain, there are seven different segments called rhombomeres (r). From the rhombomeres, the CNC cells migrate in three large currents into the pharyngeal arches (BA). (**b**) The image of the skull shows the contribution of the different CNC populations to the elements of the skull. The bones are color-coded according to the different CNC migration streams. (**c**) Most of the tissues of the teeth are derived from CNC cells (printed in black). Two exceptions are enamel and blood vessels (printed in grey). Fr = os frontal, N = os nasal, Zy = os zygomaticum, Mx = maxilla, Mn = mandibula, Sq = pars squamosa of the Os temporale, As = sphenoid wings, r = rhombomeres, BA = pharyngeal arches. Schematic drawing orientating on [6].

**Figure 2 ijms-22-01315-f002:**
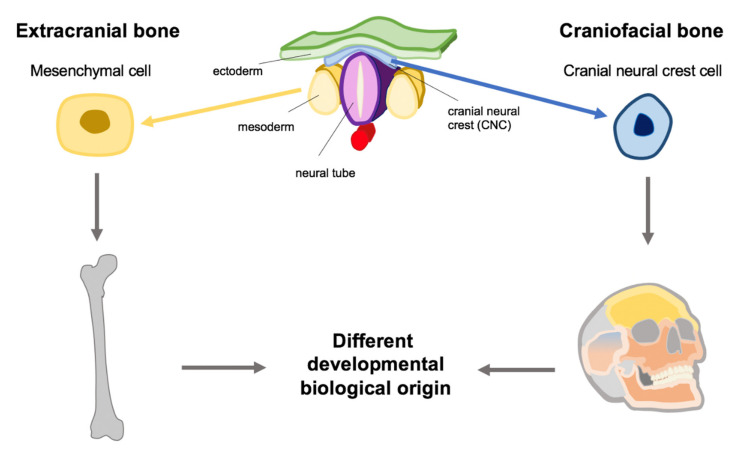
Different embryonic origin of craniofacial and extracranial bones. The different biological characteristics of craniofacial and extracranial bones can be explained by the different developmental origin (mesenchymal vs. CNC). Schematic drawing orientating on [17].

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
