# Peer review of "The Special Developmental Biology of Craniofacial Tissues Enables the Understanding of Oral and Maxillofacial Physiology and Diseases"

_ijms, 2021, doi:10.3390/ijms22031315_

Round 1
Reviewer 1 Report
- A reference must be made to differences between cartilagenous epiphysis of the long bones and mandibular condylar cartilage of skeletogenic origin with high adaptation to mechanical loading.Also the genetic involvement in the onset of temporomandibular disorders(D. Carlson 2014, JJ Mao and H.D. Nah 2004).
- It must be pointed out the significance of mechanical loading for bone growth and development and the importance of mechanical stimuli for cell differentiation, proliferation and metabolism. Also it must be pointed out that the outcome of dentofacial orthopedic appliances is mostly due to growth remodeling instead of skeletal increase due to growth stimulation(MC Meikle 2007).
- It must mentione that the skull has two distinct regions: the viscerocranium and the neurocranium. The viscerocranium is involved during feeding and breathing and its growth is dependent to muscular loading ,whereas the neurocranium contains the brain and grows with its expansion.
Author Response
see attchment.

Reviewer 2 Report
The authors present a narrative review about the embryonic development of maxillofacial hard tissues, with a potential translational insight for relevant pathologies in maxillofacial surgery, dentistry and orthodontic therapy. The text is concise and comprehensive, and the figures are really well done. However, i have some issues:
Introduction
I think that a short paraghraph explicating the aim of the present work should be indicated. The abstract shows such information, whereas in the main text body there is not.
Methods
Despite it is well written, I think that (mainly for narrative review) a methods section clarifying how the reviews was conduct, and in which manner the articles were selected for inclusion, should be necessary.
I also suggest to remove "Review" from the title, because it appears quite redundant
Round 2
Reviewer 2 Report
I think that now the article is suitable for publication